# A porous metal-organic cage liquid for sustainable $CO_2$ conversion reactions

Chang He[1,2,3], Yu-Huang Zou[1], Duan-Hui Si[1], Zi-Ao Chen[1,3], Tian-Fu Liu [1,3], Rong Cao [1,3,4] ✉ & Yuan-Biao Huang [1,3] ✉

Porous liquids are fluids with the permanent porosity, which can overcome the poor gas solubility limitations of conventional porous solid materials for three phase gas-liquid-solid reactions. However, preparation of porous liquids still requires the complicated and tedious use of porous hosts and bulky liquids. Herein, we develop a facile method to produce a porous metal-organic cage (MOC) liquid (Im-PL-Cage) by self-assembly of long polyethylene glycol (PEG)-imidazolium chain functional linkers, calixarene molecules and Zn ions. The Im-PL-Cage in neat liquid has permanent porosity and fluidity, endowing it with a high capacity of $CO_2$ adsorption. Thus, the $CO_2$ stored in an Im-PL-Cage can be efficiently converted to the value-added formylation product in the atmosphere, which far exceeds the porous MOC solid and nonporous PEG-imidazolium counterparts. This work offers a new method to prepare neat porous liquids for catalytic transformation of adsorbed gas molecules.

Porous materials are fascinating materials that are continually the focus of research and have shown enormous applications in adsorption and separation, energy storage and conversion, and heterogeneous catalysis[1-3]. Nevertheless, most porous materials are in the solid state, and their solid nature prevents their integration into the usual flow processes[4]. In addition, mechanical fatigue and physical aging problems are common occurrences with solid materials. In contrast to porous solids, porous liquids (PLs) are a new type of porous material with the characteristics of the permanent porosity of porous solids. They also exhibit fluidity, rapid transfer of heat, and mass common in liquids. These unique physicochemical properties make PLs a promising application prospect in many different fields. Although PLs were conceptualized by James et al. in 2007, the first PL was reported by the same team only in 2015[5,6]. PLs generally are in three classes: Type I has rigid molecular hosts forming neat liquids. Type II and III are porous hosts in a bulky solvent that fails to penetrate into the pores, which thus retain porosity. Types II and III are analogous to solutions and colloidal suspensions, respectively. Type I PLs have no boundary separating the porous solid materials from the external medium. They are therefore true single-phase fluid systems

and can provide access to the highest pore concentration per unit volume[7-10]. However, the construction of type I PLs is difficult due to the need to keep neat molecular hosts in a liquid form while maintaining their rigid inner cavities[11-13]. Accordingly, new strategies to fabricate type I porous liquids are desirable.

To date, a majority of the reported porous liquids have been described in terms of gas capture and separation[14-16]. It is necessary to expand the potential applications of these materials, thus greatly promoting their development. For example, research into catalysis is of ever-increasing importance in both the academic and industrial fields[17-20]. However, traditional homogeneous catalysts are limited by their recoverability after the completion of the reaction, and heterogeneous catalysts by the availability of active sites, respectively. Although porous solid catalysts have a larger number of accessible active sites for reactant molecules in gas–solid reactions, their pores are usually occupied by solvents in gas–liquid–solid three-phase reactions. Thus, the low concentration of the gas in the liquids usually leads to unsatisfactory activity. Porous liquids can improve gas solubility in catalytic systems and render a higher catalytic efficiency in gas-involved reactions. Further, due to the presence of multiple interfaces

[1]State Key Laboratory of Structural Chemistry, Fujian Institute of Research on the Structure of Matter, Chinese Academy of Sciences, 350002 Fuzhou, P. R. China. [2]College of Ecological Environment and Urban Construction, Fujian University of Technology, 350118 Fuzhou, Fujian, P. R. China. [3]University of Chinese Academy of Sciences, 100049 Beijing, P. R. China. [4]Fujian Science & Technology Innovation Laboratory for Optoelectronic Information of China, 350108 Fuzhou, Fujian, P. R. China. ✉e-mail: rcao@fjirsm.ac.cn; ybhuang@fjirsm.ac.cn

in porous liquids, it is anticipated that porous liquids could be as good as homogeneous and heterogeneous catalysts, and are more suitable for conventional industrial catalytic flow settings. Compared with gas sorption and separation, the application of PLs as catalysts is currently unusual[21–23].

Porous metal-organic cages (MOCs) are assembled from metal ions or clusters and multidentate ligands via a coordination bond and are one class of promising multifunctional materials[24–28]. Due to their discrete zero-dimensional framework structures, accessible intrinsic cavities, and unique chemical microenvironments, MOCs have shown efficient competence and selectivity of catalysis via enrichment substrates, stabilization of intermediates, and promotion region activation. Moreover, MOCs also can usually be dissolved without breaking chemical bonds by precise design, which makes them suitable for the preparation of Type 1 porous liquids through a supramolecular complexation strategy. Consequently, MOCs may be promising candidates for the fabrication of MOC-based porous liquids by providing tunable void architectures for the deployment of chemical reactions from adsorbed molecules[29–31].

Fabrication of MOC-based porous liquid catalysts, requires the prevention of the functional groups or guest molecules blocking the cavities thus lowering the melting points of such materials. In addition, the viscosity of such liquids should also be limited. It is well-known that imidazolium compounds and poly(ethylene glycols) (PEG) are moderately viscous liquid compounds[32]. Thus, surface modification of MOCs with functional groups would liquefy MOCs, thus lowering their melting points.

In this work, a facile method was devised to generate a porous MOC liquid, Im-PL-Cage in the neat state, by incorporating PEG-imidazolium chains into the periphery of the MOC (Fig. 1). As coulombic repulsion could keep the chains from entering the positively charged cavities they were terminated by positive imidazolium moieties. As a proof of concept, the PEG-imidazolium chains were integrated into a *p-tert*-butylsulfonylcalix[4]arene-based MOC to form the porous liquid Im-PL-Cage, which was self-assembled by the ionic ligand PEG-imidazolium 1,3-benzenedicarboxylic acid (PEG-Im-$H_2$BDC), Zn(NO$_3$)$_2$·6H$_2$O and a *p-tert*-butyl-sulfonylcalix[4]arene (H$_4$TBSC) via coordination bonds. The long PEG chains could not only shield guaranteeing the accessibility of the host cavities but also could lower the melting point of the Im-PL-Cage and cause it to behave as a liquid. Consequently, the porous Im-PL-Cage obtained in this way is accessible to CO$_2$ molecules and shows greatly enhanced CO$_2$ adsorption

uptake, approximately 6 times larger than that of PEG-Im-H$_2$BDC liquid. Compared with the porous MOC solid counterpart Zn-Cage lacking the PEG chains, the unique feature of this porous Im-PL-Cage is its stronger CO$_2$ storage ability and ability to act as a gas reservoir for *N*-formylation reactions with CO$_2$. To the best of our knowledge, this is the first report on the use of a porous MOC liquid as a CO$_2$ storage material for catalysis which could lead to new insights in this field.

## Results

### Synthesis and characterization of Zn-Cage

To validate our design strategy, we first prepared an imidazolium-functionalized ionic MOC without PEG chains, termed a Zn-Cage. The Zn-Cage was produced by a reaction of the ionic ligand 5-[(3-methyl-imidazol-1-yl) chloride]−1,3-benzene- dicarboxylic acid (5-Me-im-1,3-H$_2$BDC)$^+$(Cl$^-$), Zn(NO$_3$)$_2$·6H$_2$O and H$_4$TBSC (Supplementary Figs. 1−7) in a mixture of dimethyl formamide (DMF) and methanol (MeOH) at 100 °C for 24 h, which gave rise to yellow block-shaped crystals of Zn-Cage (Supplementary Fig. 8). The Zn-Cage was characterized by single crystal X-ray diffraction (SCXRD, Supplementary Figs. 9, 10, Supplementary Table 1), powder X-ray diffraction (PXRD, Supplementary Fig. 11), high-resolution mass spectrometry (MS, Supplementary Fig. 12), thermogravimetric analysis (TGA, Fig. 1d), inductively coupled plasma atomic emission spectroscopy (ICP-AES, Supplementary Table 2) and elemental analysis (EA, Supplementary Table 2). Accordingly, the formula of Zn-Cage was determined to be {[Zn$_4$(TBSC) ($\mu_4$-OH)]$_4$-(5-Me-im-1,3-BDC)$^+_8$Cl$^-_8$}(DMF)$_{10}$(CH$_3$OH)$_{15}$(H$_2$O)$_{23}$.

According to the SCXRD analysis, Zn-Cage crystallizes in the triclinic *P*ī space group. As shown in Supplementary Fig. 9, Zn-Cage consists of four tetranuclear [Zn$_4$($\mu_4$-OH)(TBSC)] units bridged by eight [(5-Meim-1,3-H$_2$BDC)$^+$(Cl$^-$)] linkers. The eight imidazolium groups of linkers are arranged around the two rims of a barrel structure and are suitable for incorporation of PEG-imidazolium chains into the periphery of the parent Zn-Cage to yield Im-PL-Cage. The imidazolium active site can be fully exposed to substrates and is thus favorable to the chemical fixation of CO$_2$. Meanwhile, the SCXRD showed that eight Cl$^-$ counteranions are positioned around the imidazolium groups. The barrel-shaped *endo* cavity of Zn-Cage has two small, 6.8 Å × 4.5 Å portals (including the van der Waals radii), which can be easily penetrated by CO$_2$ molecules but prevents the large PEG chains from entering the cavities of the cage (Supplementary Fig. 9). This is important because it shows that the empty cavities in the liquid state are still accessible to CO$_2$ gas molecules but not to PEG. The size

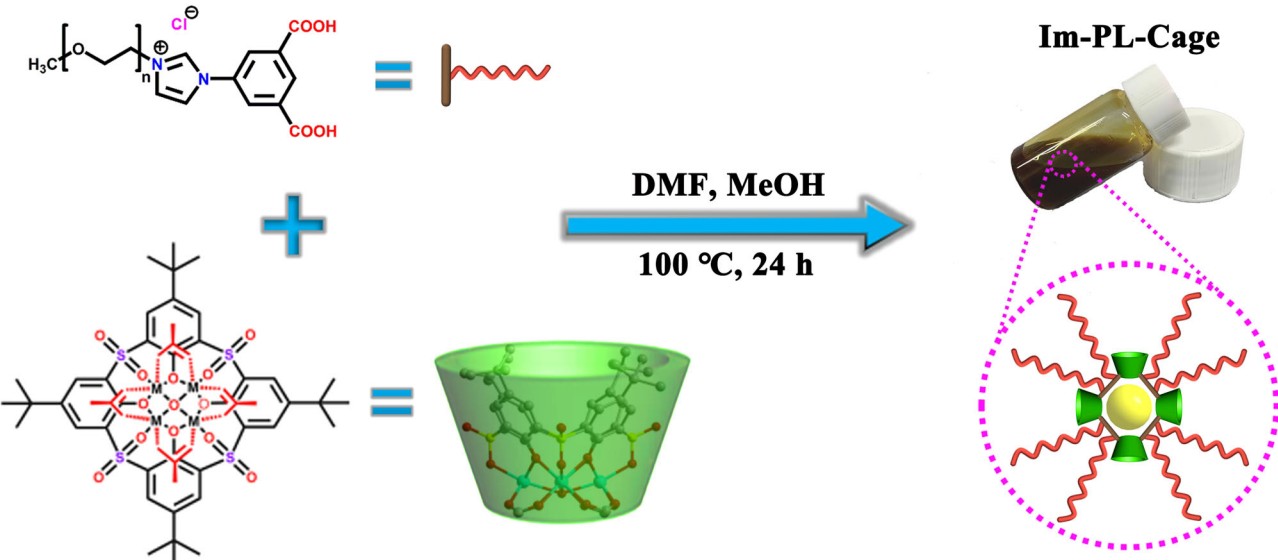

**Fig. 1 | Schematic diagram for the preparation of porous liquid Im-PL-Cage.** The self-assembly synthesis of Im-PL-Cage from PEG-Im-H$_2$BDC and Zn$_4$(TBSC).

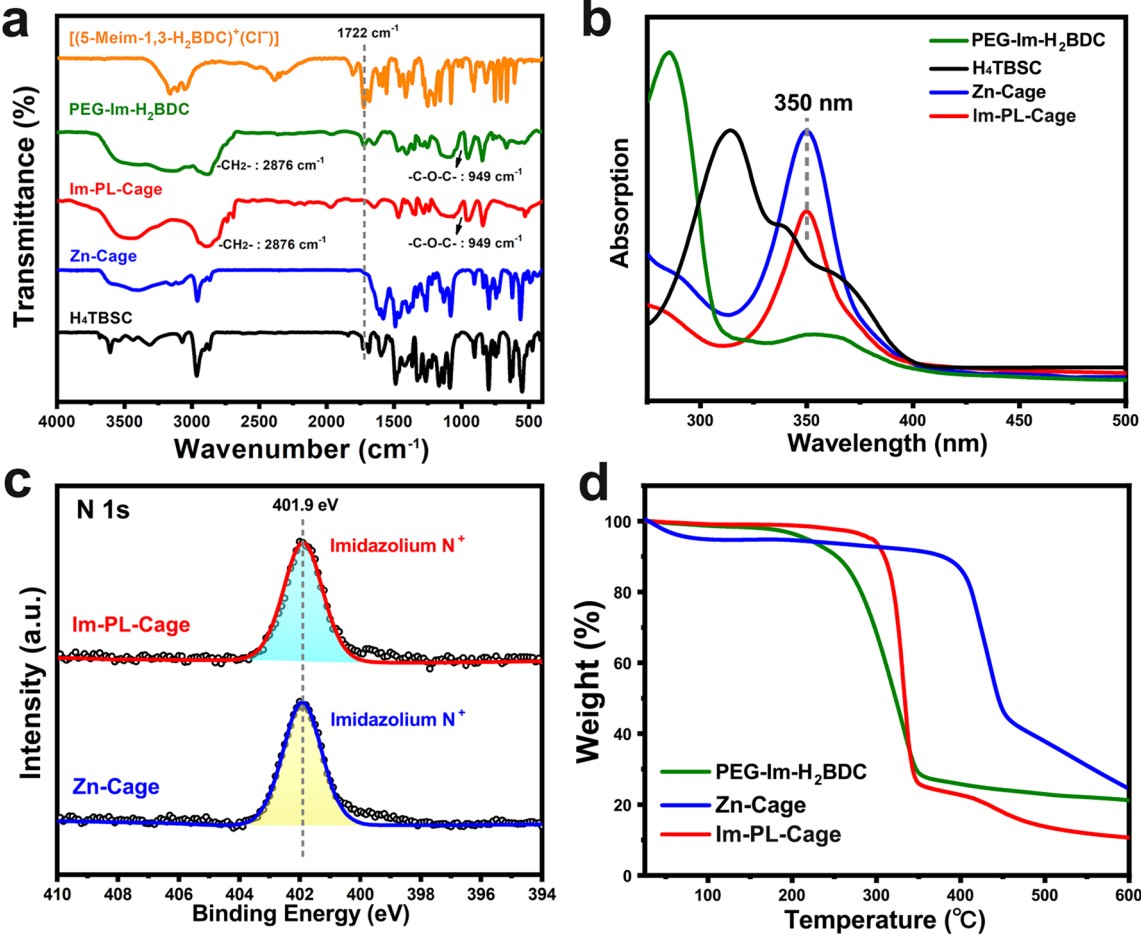

**Fig. 2 | The characterization of cages and ligands. a** FT-IR spectra of [(5-Meim-1,3-H$_2$BDC)$^+$(Cl$^-$)], PEG-Im-H$_2$BDC, Im-PL-Cage, Zn-Cage and H$_4$TBSC. **b** UV–Vis spectra of PEG-Im-H$_2$BDC, H$_4$TBSC, Zn-Cage and Im-PL-Cage. **c** XPS spectra for the N 1$s$ regions of the Im-PL-Cage and the Zn-Cage: XPS raw data (open cycles) and fitting curves (line). **d** TGA for the PEG-Im-H$_2$BDC, Zn-Cage and Im-PL-Cage.

of the inner cavity of the Zn-Cage is 20.0 × 12.9 × 12.8 Å$^3$, and it is large enough to accommodate catalytic substrates (Supplementary Figs. 9, 10). The PXRD patterns of this sample are consistent with the simulated peaks from the single crystal data, indicating the pure phase of a bulk sample of the Zn-Cage (Supplementary Fig. 11).

The formation of the Zn-Cage was consistent with the FT-IR and UV–Vis spectra. The stretching peak at 1722 cm$^{-1}$ from the protonated carboxylate groups in the FT-IR (Fig. 2a) disappeared, showing that the [(5-Meim-1,3-H$_2$BDC)$^+$(Cl$^-$)] ligands were fully coordinated with the zinc metal centers. Furthermore, the peak in the UV–Vis absorption spectra of Zn-Cage in a MeOH solution is located at 350 nm (Fig. 2b), and results from the π → π* electronic transitions of molecules with significant intramolecular charge transfer (ICT) character from TBSC units to the (5-Meim-1,3-BDC)$^-$ segments. This observation indicates the essential intactness of the cages. In addition, it is clear from the XPS that the binding energy of N 1$s$ is 401.9 eV for Zn-Cage due to the quaternary-N in the imidazolium groups (Fig. 2c). This result is consistent with the FT-IR and SCXRD data and indicates the existence of imidazolium moieties in the Zn-Cage. TGA demonstrated that the Zn-Cage is thermally stable up to 400 °C (Fig. 2d), and the slight weight loss before 100 °C, which was mainly attributed to the evaporation of residual methanol.

To confirm the porosity of Zn-Cage, the N$_2$ sorption isotherms at 77 K were measured with activated samples. As shown in Supplementary Fig. 13, the Zn-Cage has saturated N$_2$ uptakes of 152 cm$^3$ g$^{-1}$ at 1.0 bar, corresponding to the Brunauer–Emmett–Teller (BET) surface area of ~295 m$^2$ g$^{-1}$. The pore volume of Zn-Cage is 0.23 cm$^3$ g$^{-1}$ and the

micropore size is ~0.6 nm, which is in agreement with the data from the single-crystal X-ray structure. Measurements of CO$_2$ adsorption were also conducted to verify the CO$_2$ capture capability of the Zn-Cage. As depicted in Supplementary Fig. 13b, Zn-Cage exhibits high CO$_2$ adsorption uptakes with 27 cm$^3$ g$^{-1}$ at 298 K, indicating the strong affinity for CO$_2$ of this material.

## Synthesis and characterization of Im-PL-Cage

The successful synthesis of the imidazolium-functionalized cationic MOC encouraged us to prepare a Type 1 porous liquid Im-PL-Cage modified with PEG-imidazolium chains in its neat state. To retain the hollow structures in the liquid state, it is necessary to prevent the fluid medium from filling the cavities. Toward this end, the PEG-imidazolium chain functional linker (PEG-Im-BDC) ~13 nm in size and with a molecular weight ($M_w$) of ~4000 g/mol was synthesized for modification of the Zn-Cage (Figs. 1, 2a–d, and Supplementary Figs. 3, 4). Subsequently, we replaced [(5-Meim-1,3-H$_2$BDC)$^+$(Cl$^-$)] with PEG-Im-BDC, and obtained the Im-PL-Cage under synthetic conditions similar to those used for the Zn-Cage. Im-PL-Cage is a viscous paste-like material at room temperature (Fig. 1), with a melting point of ~58 °C (Fig. 5b), which makes it a liquid at readily accessible temperatures (Supplementary Movie 1). The FT-IR and UV–Vis spectra of the Im-PL-Cage that is obtained in this way are consistent with those of the Zn-Cage (Fig. 2a, b), and suggest their similar coordination structures. From the XPS curves (Fig. 2c) it can be seen that the peak of N 1$s$ at 401.9 eV is attributable to the quaternary-N of the imidazolium groups in Im-PL-Cage. The presence

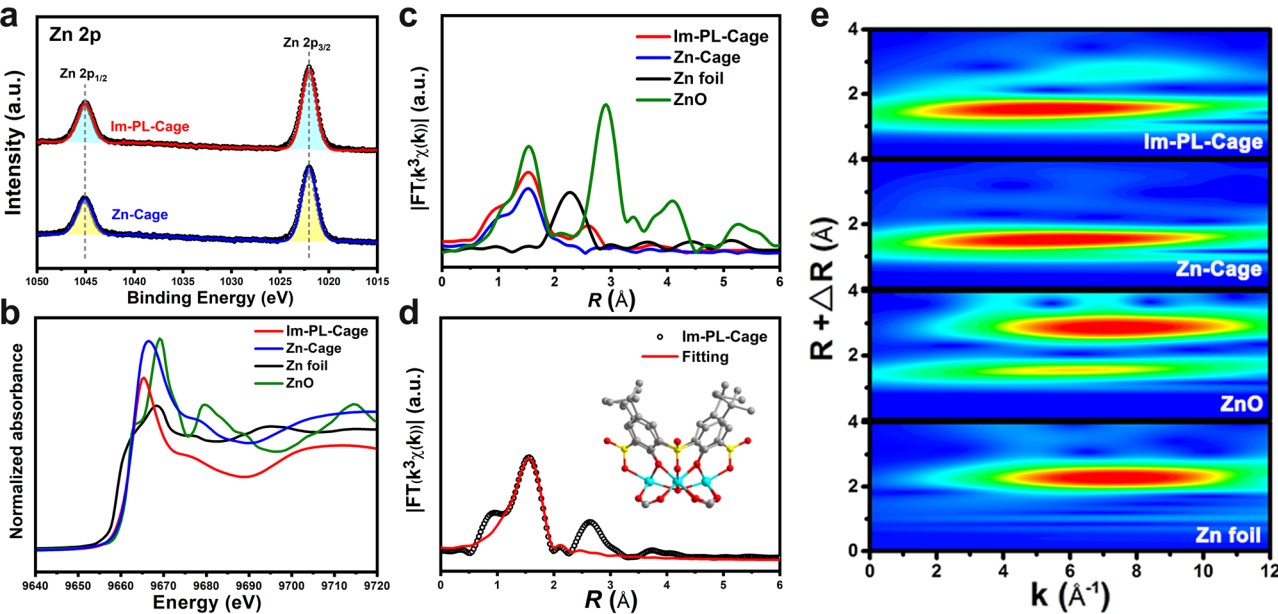

**Fig. 3 | The structural characterization of Im-PL-Cage and Zn-Cage. a** XPS spectra for the Zn 2*p* regions of Im-PL-Cage and Zn-Cage: XPS raw data (open circles) and fitting curves (line). **b** The normalized Zn K-edge XANES. **c** Fourier transform EXAFS spectra of Im-PL-Cage, Zn-Cage, Zn foil, and ZnO. **d** Fourier transform EXAFS fitting curves of Zn K-edge for Im-PL-Cage (inset: structural model of tetranuclear [$Zn_4(\mu_4$-OH)(TBSC)] unit). **e** Continuous Cauchy wavelet transform $k^3$-weighted EXAFS spectra of Im-PL-Cage, Zn-Cage, ZnO, and Zn foil.

of uncoordinated chloride anions was confirmed by the peaks of Cl 2$p_{1/2}$ and Cl 2$p_{3/2}$ at 199.2 and 197.6 eV, respectively (Supplementary Fig. 14). The high-resolution Zn 2*p* spectra in Fig. 3a have two main peaks at 1021.4 eV (2$p_{3/2}$) and 1044.6 eV (2$p_{1/2}$), which can be attributed to the formation of $Zn-O_6$ coordination structure in both Zn-Cage and Im-PL-Cage.

Synchrotron-based X-ray absorption spectroscopy (XAS) was used to confirm the coordination environment of Zn species in the Zn-Cage and the Im-PL-Cage[33–35]. As can be seen in the X-ray absorption of the near-edge structure (XANES) of the Zn K-edge in Fig. 3b, the edge positions of both of Zn-Cage and Im-PL-Cage coincide with those of ZnO but deviate from that of the Zn foil, revealing that the valence states of Zn in the two materials are close to +2. Concurrently, the local environment of the Zn species in the Zn-Cage and the Im-PL-Cage were further analyzed with the extended X-ray absorption fine structure (EXAFS). As shown in Fig. 3c, the Fourier-transformed EXAFS curves of Zn-Cage and Im-PL-Cage have main peaks at 1.53 Å, which are derived from the Zn−O bonds and are distinctly different from the main characteristic peaks of the Zn−Zn bond in Zn foil. A least-squares EXAFS fitting for the Fourier-transformed EXAFS curves were performed using a square geometry structure. It was found that the Im-PL-Cage model matched very well with the crystal structure of the Zn-Cage (Fig. 3d, Supplementary Table 3). In addition, the wavelet transform (WT) contour plots of both Zn-Cage and Im-PL-Cage show the maximum intensity at -5.8 Å$^{-1}$ (Fig. 3e), corresponding to the respective Zn−O coordination in each of the species. These results differ from that of ZnO and Zn foil, which confirms the existence of the Zn-Cage coordination structure in Im-PL-Cage and the success of our synthetic strategy.

It was seen from the TGA curves that Im-PL-Cage could be stable up to 200 °C without no obvious weight loss, indicating that no volatile solvents remained in the cavities of MOC in Im-PL-Cage (Fig. 2d). In further examination of the nanostructure, TEM and high-angle annular dark-field scanning transmission electron microscopy (HAADF-STEM) of Im-PL-Cage were performed. The TEM images of Im-PL-Cage (Fig. 4a, b) reveal its porous structure. The TEM element mapping images demonstrated that the elements Zn, S, N, and Cl are homogeneously distributed over the entire architecture. Molecular

dynamics (MD) simulations were carried out to highlight the permanent porosity of Im-PL-Cage. In the Im-PL-Cage simulation boxes, 4–6 Å pores were observed clearly, and can be easily penetrated by $CO_2$ molecules (Fig. 4c and Supplementary Fig. 15). The PEG-imidazolium chains were examined by density functional theory (DFT) calculations and, as shown in Supplementary Fig. 16, the length, width, and height of PEG-imidazolium chains functional linkers in a stable state are -132, 26 and 17 Å, respectively, showing that the linker is too large to enter the cavities of the Im-PL-Cage (Supplementary Fig. 9). Therefore, the cavities of Im-PL-Cage could remain empty in the liquid state, and the imidazolium active sites will be still accessible to $CO_2$ gas molecules during the catalytic process.

Experiments with positron ($e^+$) annihilation lifetime spectroscopy (PALS) an in-situ pore or void-volume characterization technique[36–38] was used to confirm the vacant character of the cages in the porous liquid (Fig. 5a). PALS is a procedure in which the *ortho*-positronium (o-Ps), a parallel spin complex of a positron and an electron, can be generated within low electron density regions of the insulating material, e.g. empty cavities by exposing the material to a positron source such as $^{22}$Na. The average pore diameter of the sample is correlated to the lifetime of the o-Ps because larger pores are correlated with slower decay rates. The o-Ps lifetime for the neat liquid Im-PL-Cage, determined by the measured positron lifetime spectrum is 4.85 ± 0.05 ns, which corresponds to an average void diameter of 9.34 ± 0.08 Å, implying that permanent micropores have been successfully incorporated into the liquid state.

The phase-transition behavior and rheological properties of Im-PL-Cage were investigated by differential scanning calorimetry (DSC) and analysis of the oscillation-strain moduli. As shown in Fig. 5b, an obvious first-order melting phase transition appeared at 58 °C, and corresponds to the melting temperature of Im-PL-Cage. Notably, the corresponding melting temperature for PEG-imidazolium chains is lower than that from the Im-PL-Cage. This could be a result of the introduction of cationic Zn-based metal-organic polyhedrons, which could produce locally ordered regions such as liquid crystals. Consequently, more energy and higher temperatures are necessary for Im-PL-Cage to break the ionic bonds of the imidazolium moieties, suggesting that the Im-PL-Cage was more stable than the system

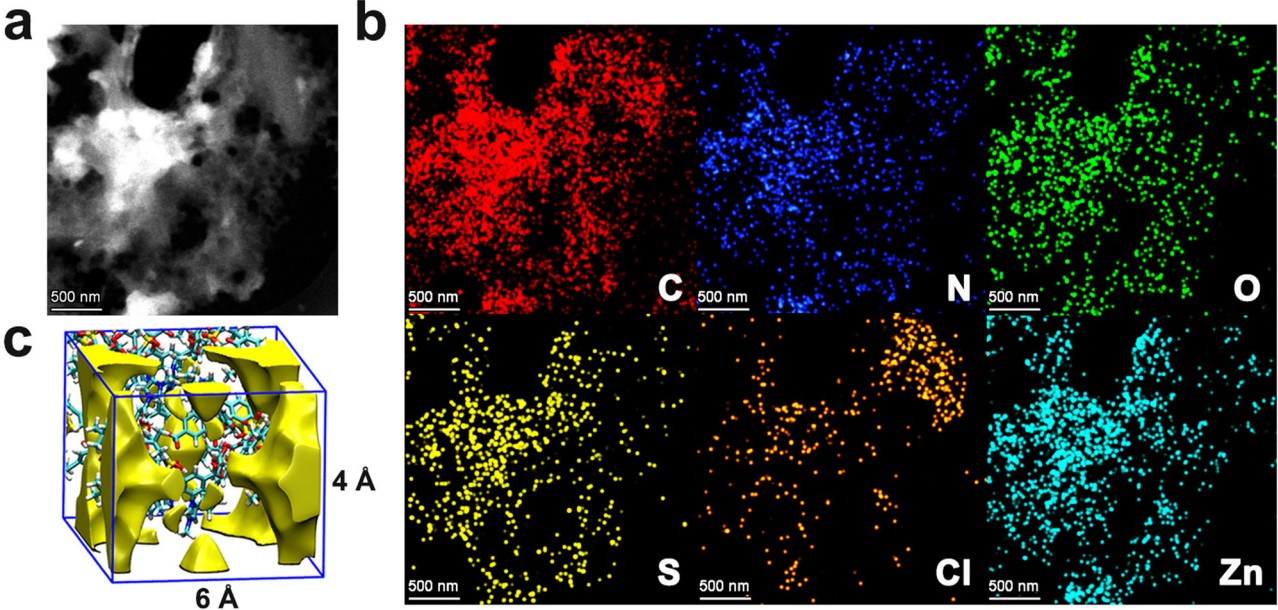

**Fig. 4 | The Morphology and structure of Im-PL-Cage. a** HAADF-STEM image, **b** the corresponding EDX mapping of Im-PL-Cage, and **c** snapshots of simulation boxes for Im-PL-Cage and pore space in Im-PL-Cage.

involving pure PEG-imidazolium chain functional linkers. In further investigation of the fluidity of Im-PL-Cage, the oscillation-strain modulus was measured and, as shown in Fig. 5c, the loss modulus ($G'$) was consistently higher than the storage modulus ($G''$) throughout the measured oscillation range, indicating that Im-PL-Cage exhibits liquid behavior at 60 °C.

### Im-PL-Cage for CO$_2$ adsorption and catalytic conversion

Such a porous liquid material may offer many opportunities for specific applications, such as gas storage and catalysis. To further examine and utilize the permanent cavities in porous liquids, gravimetric gas solubility measurements of CO$_2$ in Im-PL-Cage and PEG-Im-BDC were performed at ambient temperature. As shown in Fig. 5d, pronounced enhancement of the CO$_2$ adsorption capacity (1.78 mmol g$^{-1}$, 10 bar) was detected in the case of Im-PL-Cage in comparison with the PEG-Im-BDC (0.31 mmol g$^{-1}$, 10 bar). The higher CO$_2$ adsorption capacity of Im-PL-Cage further indicated that the porous liquid has permanent pores which provide additional free volume for storage of CO$_2$ molecules[39–41]. Benefiting from the presence of permanent porosity, porous liquids are more able to adsorb and enrich gas. The literature on most porous liquids is focused on the applications of gas separation and storage, and the development of their use as catalysts for gas stored in porous liquids is expected to be developed.

An important way to recycle carbon dioxide is to combine the CO$_2$ reduction and construct a C–C, C–N, or C–O bond thus producing value-added chemicals. For instance, reductive functionalization of CO$_2$ with amines and phenylsilane to afford formamides selectively has attracted increasing attention[42–45]. Formamides are versatile chemicals and important building blocks, generally produced by the formylation of amines. Use of CO$_2$ rather than toxic CO for the N-formylation process is an attractive and green alternative for the production of formamides. The imidazolium-based ionic liquids (ILs) are one of the catalysts used for the reaction of CO$_2$ with amines and phenylsilane to form formamides[42], but their activities are usually limited by the low CO$_2$ solubility. Inspired by the good CO$_2$ adsorption capacity of Im-PL-Cage with its unique chemical structure and pore environment, we investigated the use of CO$_2$ stored in the Im-PL-Cage as a gas source for the reductive formylation reactions. Thus, the stored CO$_2$ in the pores of Im-PL-Cage can be easily accessed and activated with the nearby

imidazolium sites on the wall of cages and subsequent reaction with a substrate amine. For this purpose, we designed a two-step experiment. In the first step, CO$_2$ was adsorbed and stored in Im-PL-Cage in an autoclave containing CO$_2$ at 20 bar until the maximum uptake was reached. The valve of the autoclave was then opened until the internal pressure and the atmospheric pressure were balanced. In the second step, the amines and phenylsilane were added to the autoclave to react with the CO$_2$ that had been stored and then released from Im-PL-Cage at 60 °C in an air atmosphere.

The formylation reaction of morpholine with CO$_2$ and phenylsilane was chosen for a proof-of-concept using the CO$_2$ stored in a porous MOC liquid system (Fig. 6). In a control experiment, only morpholine and phenylsilane were added into the autoclave at 60 °C in an air atmosphere in the absence of any adsorbent. As shown in Fig. 6 and Supplementary Table 4, under these conditions, only 1.63 mmol of morpholine was transformed into N-formylmorpholine in a 16.3% yield (Supplementary Table 4, entry 1), which was produced from the CO$_2$ remaining in the autoclave and the pipeline. When PEG-Im-BDC was employed as a CO$_2$ adsorbent, N-formylmorpholine was obtained in 21.5% yield and the amount of CO$_2$ adsorbed on PEG-imidazolium chain was calculated to be 0.52 mmol (Fig. 6, Supplementary Table 4, entry 2), which was close to that in the control experiment. This showed that PEG has a weak capability to adsorb CO$_2$, and was confirmed by gravimetric gas solubility measurements of CO$_2$ (Fig. 5d). In contrast, a high concentration of CO$_2$ (7.74 mmol) from in Im-PL-Cage was converted to a formamide with a high yield of 96.7% (Supplementary Table 4, entry 3). This activity was achieved because the Im-PL-Cage has permanent pores which can store CO$_2$ in a quantity almost 15 times more than that of pure PEG-Im-H$_2$BDC when both were charged at 20 bar. Notably, although the porous Zn-Cage solid showed excellent CO$_2$ adsorption uptake capacity (Supplementary Fig. 13b), only 2.45 mmol of morpholine was converted with a low yield of 24.5% (Supplementary Table 4, entry 4). This result was attributed to the fact that most of the CO$_2$ adsorbed in the cavities of the solid Zn-Cage was quickly released during decompression, and subsequently, the solvent and reaction substrates can quickly occupy the pore cavities of Zn-Cage, leading to the further escape of CO$_2$. In contrast to the Zn-Cage solid, the CO$_2$ molecules could be adsorbed and stored in the cavities of the porous liquid Im-PL-Cage longer than in an air atmosphere.

**Fig. 5 | The porosity and rheology of Im-PL-Cage. a** Positron annihilation lifetime spectroscopy of Im-PL-Cage. **b** DSC curves of PEG-Im-H$_2$BDC (green) and Im-PL-Cage (red). **c** Oscillation-dependent modulus plots of Im-PL-Cage (filled circles for storage modulus $G'$, open circles for loss modulus $G''$). **d** CO$_2$ adsorption-desorption isotherms of PEG-Im-H$_2$BDC (green) and Im-PL-Cage (red) at ambient temperature.

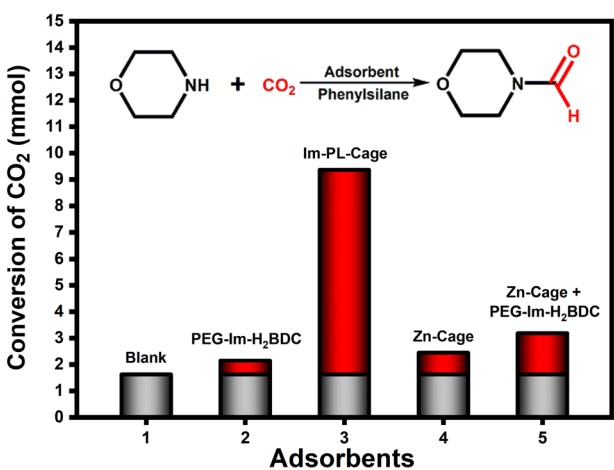

**Fig. 6 | The formylation reaction of morpholine with CO$_2$ and phenylsilane.** The experimental results of catalytic conversion of CO$_2$ to N-formylmorpholine with different adsorbents (gray bars represent the transformed amount of CO$_2$ that occupied the reactor and red bars represent the transformed amount of CO$_2$ that was adsorbed on adsorbents).

Besides, the mixture of Zn-Cage and PEG-Im-BDC also performed poorly in this reaction delivering a yield of 31.9%, and the amount of captured CO$_2$ was determined to be 1.56 mmol (Supplementary Table 4, entry 5), indicating that simple physical mixing of Zn-Cage with PEG-imidazolium chains failed to afford a porous liquid which can adsorb and store CO$_2$. These results demonstrated that Im-PL-Cage has a much better capability of enhanced CO$_2$ adsorption than Zn-Cage or PEG-Im-BDC, and this was attributed to the unique properties of a porous liquid. Encouraged by these results, we further explored the substrate scope of Im-PL-Cage. Good to excellent conversion rates were achieved, indicating that the catalyst has high substrate tolerance to this reaction (Supplementary Table 5). Notably, Im-PL-Cage not only has the merits of a homogeneous catalyst, such as fluidity and high solubility but also combines the recoverable nature of heterogeneous ones, so the reusability of Im-PL-Cage can also contribute to the results. In each cycle, the Im-PL-Cage was removed by centrifugation and then washed with DMF and used in a subsequent run. Im-PL-Cage showed good reusability, which was confirmed by the fact that a 93% yield of N-formylmorpholine was achieved even after the catalyst had been reused five times with stable dispersion and assembly (Supplementary Fig. 17). Besides, XPS and XAS show that the Cage structure of Im-PL-Cage is well preserved after catalysis (Supplementary Fig. 18,

Supplementary Table 3). The Im-PL-Cage with unique properties can combine the easy recyclability and porosity of heterogeneous systems with the rapid mass transfer and fluidity in the homogeneous counterparts, thus providing extra opportunities in catalysis. These findings could lead to practical applications in industry. For example, porous liquids could absorb gases like $CO_2$ and be subsequently pumped into a reactor for the appropriate reactions.

## Discussion

We report the design and preparation of Im-PL-Cage, a new Type I porous liquid metal-organic cage, by the introduction of a high-density of PEG-imidazolium chain functional linkers into a calix[4]arene-based MOC. The long PEG-imidazolium chains can serve as coronas on the surface of Im-PL-Cage which cause the cavities of Im-PL-Cage to remain empty in the neat liquid state. The permanent porosity in the Im-PL-Cage was confirmed by PALS and $CO_2$ sorption measurements and a high concentration of $CO_2$ can be stored in its cavities and can be released and converted to formylation products with high yields. The Im-PL-Cage can be recycled at least five times without obvious loss of activity, confirming its recyclability and stability. This work provides a proof of concept for the deployment of chemical reactions within porous liquids and the development of useful chemicals from adsorbed molecules. This can advance the concepts and prospects of PLs and narrow the gaps between homogeneous and heterogeneous systems.

## Methods

### Synthesis of Zn-Cage
$Zn(NO_3)_2 \cdot 6H_2O$ (74.4 mg, 0.25 mmol), [(5-Meim-1,3-$H_2$BDC)$^+$(Cl$^-$)] (41.1 mg, 0.11 mmol) and TBSC (42.5 mg, 0.05 mmol) were dissolved in 10 mL of *N,N*-dimethylformamide (DMF) and 5 mL of methanol in a scintillation vial (20 mL capacity). The vial was placed in a sand bath, which was transferred to a programmable oven and heated at a rate of 0.5 °C/min from 35 to 100 °C. The temperature was held at 100 °C for 24 h before the oven was cooled at a rate of 0.2 °C/min to a final temperature of 35 °C. Yellow crystals of Zn-Cage that formed in 5 days were isolated by washing with methanol and dried in the air to give 63.5 mg of the as-synthesized material. The cif can be found in https://www.ccdc.cam.ac.uk/ with the ccdc number 2259420.

### Synthesis of Im-PL-Cage
$Zn(NO_3)_2 \cdot 6H_2O$ (74.4 mg, 0.25 mmol), PEG-Im-BDC (473 mg, 0.11 mmol) and TBSC (42.5 mg, 0.05 mmol) were dissolved in 10 mL of *N,N*-dimethylformamide (DMF) and 5 mL of methanol in a scintillation vial (20 mL capacity). The vial was placed in a sand bath, which was transferred to a programmable oven and heated at a rate of 0.5 °C/min from 35 to 100 °C. The temperature was held at 100 °C for 24 h before the oven was cooled at a rate of 0.2 °C/min to a final temperature of 35 °C. The cage was precipitated out of solution via the addition of excess diethyl ether and centrifuged. The resulting oil was concentrated under reduced pressure yielding a dark yellow, viscous liquid (378 mg).

### Details of the hydrosilylation of $CO_2$ to formamides
In a typical procedure, the Im-PL-Cage (300 mg) was placed in an autoclave at 298 K. After being sealed, the autoclave was purged thrice with $CO_2$ and filled with 20 bar $CO_2$. Subsequently, the autoclave was rapidly frozen by liquid nitrogen for 3 min. Excess pressure was released to 1 bar, opening the autoclave to exclude the rest $CO_2$. Afterward, the morpholine (0.87 mL, 10 mmol), phenylsilane (2.47 mL, 20 mmol), and 5 ml DMF were added to the autoclave. After being sealed again, the reaction was carried out at 60 °C for 10 h. After the reaction was completed, small amounts of the mixture were withdrawn, diluted with ethanol, and filtered with a 0.22 μm membrane filter. The product was determined by gas chromatography. All the control experiments were carried out under the same reaction conditions with different carbon dioxide adsorbents.

## Data availability
The authors declare that all data supporting the findings of this study are included within the article and its Supplementary Information, which is also deposited in figshare repository (https://doi.org/10.6084/m9.figshare.22731656) and is also available from the authors upon request. The X-ray crystallographic coordinates for structures reported in this study have been deposited at the Cambridge Crystallographic Data Centre (CCDC), under deposition number 2259420. These data can be obtained free of charge from The Cambridge Crystallographic Data Centre via www.ccdc.cam.ac.uk/structures. All the calculations were performed by Gaussian 16, Revision A.03. Source data are provided with this paper.

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

## Acknowledgements

We acknowledge the financial support from the National Key Research and Development Program of China (2018YFA0208600 and 2018YFA0704502 to Y.-B.H.), NSFC (U22A20436 and 22071245 to Y.-B.H., 22033008 and 22220102005 to R.C.), and Fujian Science & Technology Innovation Laboratory for Optoelectronic Information of China (2021ZZ103 to R.C.). We thank the beamline BL14W1 station for XAFS measurements at the Shanghai Synchrotron Radiation Facility, China.

## Author contributions

C.H., Y.-H.Z., Y.-B.H., and R.C. conceived and designed the experiments. Y.-H.Z. designed the ligand synthesis. C.H. and Y.-H.Z. performed the synthetic work. C.H. and Y.-H.Z. conducted and analyzed the measurements. C.H. performed the $CO_2$ catalysis measurements. D.-H.S. and Z.-A.C. performed structure simulations. T.-F.L. discussed the idea. All the authors contributed to the manuscript preparation.

## Competing interests

The authors declare no competing interests.
