## [Peer Review File · Nature Communications]

A Porous Metal-Organic Cage Liquid for Sustainable CO₂ Conversion ReactionsReviewers' Comments:

Reviewer #1:

Remarks to the Author:

As an emerging new porous material, porous liquids combine the fluidity of liquids with the permanent porosity of porous solids, which makes them promising for various applications. In this work, Cao et al. report a facile strategy for the fabrication of a metal-organic cage (MOC) based porous liquid (Im-PL-Cage) by self-assembly method. The obtained Im-PL-Cage with a neat liquid state has permanent porosity and fluidity, endowing it with high CO₂ adsorption capacity. Interestingly, the stored CO₂ in Im-PL-Cage can be efficiently converted to the value-added formylation product in the atmosphere, which far exceeded the porous MOC solid and nonporous PEG-imidazolium counterparts. This work provides a proof of concept of deploying chemical reactions within porous liquids for developing utility chemicals from adsorbed molecules, which will be of interest to many chemists working in the catalytic field. Therefore, I suggest that this work can be accepted for publication in Nature Communications after the following minor issues are addressed.

1. The fluidity of Im-PL-Cage is not reflected in this manuscript, and the author should add corresponding pictures or videos.
2. Figure 3c, this picture should be presented more clearly.
3. Is there any explanation why the storage of CO₂ in this porous MOC liquid could facilitate CO₂ conversion? What is the function of Im-PL-Cage host as a catalyst?
4. The authors should provide some additional characterization to prove that structural composition of Im-PL-Cage is still remained after the end of catalysis.

Reviewer #2:

Remarks to the Author:

Porous liquids have great potential as new and useful materials for gas-liquid-solid reactions. However, their preparation is often challenging as it requires the use of porous hosts and bulky liquids. In this manuscript, the authors present a new method for producing a porous metal-organic cage (MOC) liquid called Im-PL-Cage. They achieved this by using self-assembly of polyethylene glycol (PEG)-imidazolium chain functional linkers, calixarene molecules, and Zn ions. Im-PL-Cage is both permanently porous and fluid, making it capable of adsorbing CO₂ and producing higher densities than conventional solvents. The authors also report N-formylation using Im-PL-Cage and CO₂. Overall, this is an interesting study that offers a new approach for preparing porous liquids for catalytic transformations. However, the study lacks some key characterizations. For instance, there is no elemental analysis. Furthermore, in my experience, imidazolium compounds with benzoic acid can produce many impurities during synthesis. Therefore, it is crucial to fully characterize all new compounds, including their color. There is a typo: accessibility.

Reviewer #3:

Remarks to the Author:

Cao, Huang, and coworkers describe the formation of porous liquid based metal organic cage and its applications in the field of CO₂ absorption due to its permanent porosity and fluidity. The study is unique focusing on intricated design and synthesis of porous metal organic cage liquid-based materials and their applications in capturing CO₂ with a target of achieving formylations product in the atmosphere. Overall, the work is well carried out and of broad interest, and may be suitable for publication in Nature Communications after revision. I do have one significant concern that affects the merit, as highlighted in points 1 and 5.

- 1) Based on this reviewer's knowledge, Type I porous liquids are very rare. The strategy invoked by the authors is likely the strategy needed to achieve these types of materials and they should be

commended for the design strategy. However, the obtained material is not a liquid at room temperature. The melting point is 58 C. While this is certainly lower than other porous solids (e.g. MOFs), I am not sure this qualifies as a liquid. Based on the data, once the solid is converted to its liquid phase via heating, it does retain porosity. The authors should provide some references on what the cutoff is for defining a porous solid as a liquid. Otherwise, many materials could be termed 'liquids' if they are just used in their melted state by heating.

2) Introduction- Delete the word in or by within at the end of this sentence: 'They also exhibit the fluidity, rapid transfer of heat and mass common in by liquids'. Only one word before liquids is needed.

3) SCXRD analysis- The authors state that there are 8 Cl⁻ anions that are near the imidazolium groups. These Cl ions are shown in Figure S7, but not in Figure S8. Most importantly, they are not present in the supplied cif file. Moreover, the supplied cif also includes additional methylimidazolium molecules that sit in two of the four calixarene cavities. The source of the methylimidazolium is not clear, and it is not shown in the SI figures. If the supplied cif is incorrect, the correct file should be supplied to the CCDC.

4) The TGA of the Zn cage shows a slight mass loss at the beginning (before 100 C). The authors should comment on the source of the mass loss. Is it unbound methylimidazolium? This type of mass loss is not seen for the PEG cage.

5) The authors state that 'Im-PL-Cage is a viscous paste-like material at room temperature (Scheme 1), with a melting point of ~50 °C (Figure 4b), which makes it a liquid slightly above room temperature.' I would argue that the use of the word slightly is a bit exaggerated as 20-25 C is room temperature. This is a low melting point solid. Moreover, DSC shows 58 C as the melting point, which is not ~50, it would be ~60 if you are rounding.

6) In the SI, the percent yields are only included for some of the synthetic steps but should be added for all.

The point-to-point responses to the referees' comments:

Reviewer: 1

As an emerging new porous material, porous liquids combine the fluidity of liquids with the permanent porosity of porous solids, which makes them promising for various applications. In this work, Cao et al. report a facile strategy for the fabrication of a metal-organic cage (MOC) based porous liquid (Im-PL-Cage) by self-assembly method. The obtained Im-PL-Cage with a neat liquid state has permanent porosity and fluidity, endowing it with high CO₂ adsorption capacity. Interestingly, the stored CO₂ in Im-PL-Cage can be efficiently converted to the value-added formylation product in the atmosphere, which far exceeded the porous MOC solid and nonporous PEG-imidazolium counterparts. This work provides a proof of concept of deploying chemical reactions within porous liquids for developing utility chemicals from adsorbed molecules, which will be of interest to many chemists working in the catalytic field. Therefore, I suggest that this work can be accepted for publication in Nature Communications after the following minor issues are addressed.

Response: Thanks for the reviewer giving us the positive comments and encouraging us to focus on this field. And we have revised carefully according to the reviewer's comments in the following.

1. The fluidity of Im-PL-Cage is not reflected in this manuscript, and the author should add corresponding pictures or videos.

Answer: Following the reviewer's professional advice, we have supplemented an additional video to illustrate the fluidity of Im-PL-Cage.

Corresponding revision in Page 5:

A facile method was devised to generate a porous MOC liquid, Im-PL-Cage in the neat state, by incorporating PEG-imidazolium chains into the periphery of the MOC

(Scheme 1 and Supporting Information Video).

2. Figure 3c, this picture should be presented more clearly.

Answer: Thank you for your careful review. We have provided several larger and sharper pictures in Supporting Information.

Corresponding revision in Page 12:

In the Im-PL-Cage simulation boxes, 4 - 6 Å pores were observed clearly, and can be easily penetrated by CO₂ molecules (Figures 3c and Figure S13).

Corresponding revision in SI:

Figure S13. The snapshots of simulation boxes for Im-PL-Cage and pore space in Im-PL-Cage. The views of the (a) X, (b) Y and (c) Z axes, respectively.

3. Is there any explanation why the storage of CO₂ in this porous MOC liquid could facilitate CO₂ conversion? What is the function of Im-PL-Cage host as a catalyst?

Answer: Compared with solid materials where most of their pores are occupied by solvents in the gas-solid-liquid system, the MOC liquid could be served as gas reserve to store CO₂ in its pores, which can easily contact with active site in the porewalls. Thus, the storage of CO₂ in this porous MOC liquid could facilitate CO₂ conversion. As for the function of Im-PL-Cage host as a catalyst, the imidazolium-based ionic liquids (ILs) are one of the most active catalysts used for the reaction of CO₂ with amines and phenylsilane to form formamide. The ILs could acted as bifunctional catalysts, which activated the Si-H bond of phenylsilane to react with CO₂ to form the formoxysilane intermediate and simultaneously activated the amine substrate through

the hydrogen bond. Moreover, the imidazolium cation and the anions of the ILs have excellent synergistic effect on catalyzing the formylation of amines (ACS Catal. 2015, 5, 4989-4993.; Chem. Eur. J. 2018, 24, 16588-16594).

4. The authors should provide some additional characterization to prove that structural composition of Im-PL-Cage is still remained after the end of catalysis.

Answer: Thanks for the valuable comment, we have supplemented the characterization of Im-PL-Cage after cyclic catalysis. As shown in Figure S16, the XPS and XAFS characterization also revealed that the coordination environment of Im-PL-Cage before and after catalysis did not change significantly. The above results indicate that Im-PL-Cage has good recyclability.

Corresponding revision in Page 19:

Besides, XPS and XAS show that the Cage structure of Im-PL-Cage is well preserved after catalysis (Supplementary Fig. 16, Supplementary Table 3).

Corresponding revision in SI:

Figure S16. (a) The XPS Zn 2p spectra of Im-PL-Cage and Im-PL-Cage (Recovered). (b) The normalized Zn K-edge XANES spectra of Im-PL-Cage and Im-PL-Cage (Recovered). (c) Fourier transform EXAFS fitting curves of Zn K-edge for Im-PL-Cage and Im-PL-Cage (Recovered).

Table S3. Fitting results from EXAFS analysis of Im-PL-Cage.

Sample	Path	CN	R(Å)	$\sigma^2(10^{-3}\text{Å}^2)$	R factor
Im-PL-Cage	Zn-O	6	2.03	4.7±1.2	0.005

(Recovered)					
-------------	--	--	--	--	--

Reviewer: 2

Porous liquids have great potential as new and useful materials for gas-liquid-solid reactions. However, their preparation is often challenging as it requires the use of porous hosts and bulky liquids. In this manuscript, the authors present a new method for producing a porous metal-organic cage (MOC) liquid called Im-PL-Cage. They achieved this by using self-assembly of polyethylene glycol (PEG)-imidazolium chain functional linkers, calixarene molecules, and Zn ions. Im-PL-Cage is both permanently porous and fluid, making it capable of adsorbing CO₂ and producing higher densities than conventional solvents. The authors also report N-formylation using Im-PL-Cage and CO₂. Overall, this is an interesting study that offers a new approach for preparing porous liquids for catalytic transformations. However, the study lacks some key characterizations. For instance, there is no elemental analysis. Furthermore, in my experience, imidazolium compounds with benzoic acid can produce many impurities during synthesis. Therefore, it is crucial to fully characterize all new compounds, including their color. There is a typo: accessibility.

Response: We are grateful to the reviewer for the positive comments and valuable suggestions to improve the quality of our research work. The test data for elemental analysis (EA) and inductively coupled plasma atomic emission spectroscopy (ICP-AES) are summarized in Supplementary Table 2. We have done our best to describe our synthesis process in detail, and we have supplemented with NMR characterization data for all the new compounds as well as their corresponding colors. We have checked the manuscript and corrected the corresponding typos.

Corresponding revision in SI:

2.1 Synthesis of [(5-Meim-1,3-H₂BDC)⁺(Cl⁻)]^{1,2}:

(1) Synthesis of dimethyl 5-(1H-imidazol-1-yl)-isophthalate

.....The product was purified by chromatography on silica gel with hexane/ethyl acetate (2/1) to yield dimethyl 5-(1H-imidazol-1-yl)-isophthalate as a yellow powder (2.362 g, 9 mmol, 18% yield). ¹H NMR (400 MHz, DMSO-*d*₆, ppm): δ = 8.26 (s, 2H), 7.97 (s, 1H), 7.39 (s, 1H), 7.27 (s, 1H), 4.00 (s, 6H).

(2) Synthesis of [(5-Meim-1,3-H₂BDC)⁺(Cl⁻)]

.....The obtained residue was rinsed with hexane/ethyl acetate (2/1) to give dimethyl 5-(3-methyl-imidazol-1-yl)-isophthalate as a brown powder (1.58 g, 3.9 mmol, 56% yield). ¹H NMR (400 MHz, DMSO-*d*₆, ppm): δ 9.81 (s, 1H), 8.38 (s, 1H), 8.20 (m, 3H), 7.85 (s, 1H), 3.88 (s, 6H), 3.94 (s, 3H).

.....The mixture was cooled down to room temperature, and the solvent was removed under reduced pressure to result [(5-Meim-1,3-H₂BDC)⁺(Cl⁻)] in light yellow powder. (0.8 g, 2.84 mmol, 73% yield). ¹H NMR (400 MHz, DMSO-*d*₆, ppm): δ 9.95 (s, 1H), 8.58 (s, 1H), 8.50 (m, 3H), 7.97 (s, 1H), 3.94 (s, 3H).

2.2 Synthesis of PEG-imidazolium 1,3-benzenedicarboxylic acid (PEG-Im-BDC):

(1) Synthesis of iodinated-methoxypolyethylene glycols

.....The desired product 4-methylbenzenesulfonate-methoxypolyethylene glycols was obtained as a white wax/oil (17.38 g, 4.1 mmol, 41% yield). ¹H NMR (400 MHz, DMSO-*d*₆, ppm): δ 7.85 (m, 2H), 7.40 (m, 2H), 3.66 (m, 360H), 3.24 (s, 3H), 2.35 (s, 3H).

.....The product iodinated-methoxypolyethylene glycols was obtained as a yellow wax/oil (15.22 g, 3.7 mmol, 90% yield). ¹H NMR (400 MHz, DMSO-*d*₆, ppm): δ 3.62 (m, 360H), 3.24 (s, 3H).

(2) Synthesis of PEG-Im-BDC

.....The final product PEG-Im-BDC as a dark yellow wax/oil (1.37 g, 0.32 mmol, 13% yield) was obtained by recrystallization from the filtrate at 273K. ¹H NMR (400 MHz, DMSO-*d*₆, ppm): δ 13.87 (s, 2H), 9.99 (s, 1H), 8.54 (m, 4H), 8.03 (s, 1H), 3.51 (m,

360H), 3.24 (s, 3H).

2.3 Synthesis of *p*-tert-butylsulfonylcalix[4]arenes (H₄TBSC)³⁻⁵:

(1) Synthesis of *p*-tert-Butylthiacalix[4]arene

.....The precipitate was collected by filtration, recrystallized from chloroform and dried in vacuo (100 °C, 4 h) to give an essentially pure sample of *p*-tert-Butylthiacalix[4]arene as pink powder (37.9 g, 49% based on the *p*-tert-butylphenol). The mother liquor of the recrystallization was concentrated in vacuo, and chromatography of the residue on silica gel (hexane/CHCl₃ = 4:6) afforded additional *p*-tert-Butylthiacalix[4]arene (3.9 g, 5%), the combined yield of *p*-tert-Butylthiacalix[4]arene amounting to 54% yield (41.8 g, 0.058 mol). ¹H NMR (400 MHz, CDCl₃, ppm): δ 9.60 (s, 4H), 7.64 (s, 8H), and 1.22 (s, 36H).

(2) Synthesis of *p*-tert-Butylsulfonylcalix[4]arene

.....The produce was recrystallized from benzene-methanol and dried in vacuo (70 °C, 12 h) to give the off-white product of *p*-tert-Butylsulfonylcalix[4]arene (1.06 g, 1.25 mmol, 90.6% yield). ¹H NMR (400 MHz, CDCl₃, ppm): δ 7.99 (s, 8H), 1.25 (s, 36H).

Reviewer: 3

Cao, Huang, and coworkers describe the formation of porous liquid based metal organic cage and its applications in the field of CO₂ absorption due to its permanent porosity and fluidity. The study is unique focusing on intricate design and synthesis of porous metal organic cage liquid-based materials and their applications in capturing CO₂ with a target of achieving formylations product in the atmosphere. Overall, the work is well carried out and of broad interest, and may be suitable for publication in Nature Communications after revision. I do have one significant concern that affects the merit, as highlighted in points 1 and 5.

Response: Thank you for your valuable suggestions to improve the quality of our

research work. And we have revised carefully according to the reviewer's comments in the following.

1. Based on this reviewer's knowledge, Type I porous liquids are very rare. The strategy invoked by the authors is likely the strategy needed to achieve these types of materials and they should be commended for the design strategy. However, the obtained material is not a liquid at room temperature. The melting point is 58 °C. While this is certainly lower than other porous solids (e.g. MOFs), I am not sure this qualifies as a liquid. Based on the data, once the solid is converted to its liquid phase via heating, it does retain porosity. The authors should provide some references on what the cutoff is for defining a porous solid as a liquid. Otherwise, many materials could be termed 'liquids' if they are just used in their melted state by heating.

Answer: Thank you for your professional advice. We think that the term 'Porous liquids' should be a holistic concept. The biggest difference between melted solids and PLs is whether there is chemical decomposition of the materials. For many conventional MOCs and MOFs, when heated to a critical temperature, they decompose due to the instability of their organic ligands and coordination bonds at high temperatures, and more importantly, these melted materials tend to be non-porous (Nat. Mater. 2017, 16, 1149-1155.). Some melted MOFs could be regarded as molten salts with unique network structures, such materials are often referred to as MOF glasses (Science. 2020, 367, 1473-1476.; J. Am. Chem. Soc. 2016, 138, 10818-10821.). PLs were conceptualized by James et al. in 2007 (Chem. Eur. J. 2007, 13, 3020-3025). In contrast to the small, transient cavities that exist between the molecules of any liquids, PLs have permanent, well-defined, empty pores capable of molecular recognition when exposed to other species (e.g., gases etc.). For Type I porous liquids, there is currently no definite limit on their melting point, accessing rigid cage structures in neat liquid form at readily accessible temperatures could be termed as PLs (Chem. Sci. 2012, 3, 2153-2157.; J. Am. Chem. Soc. 2020, 142, 7270-7275.).

2. Introduction-Delete the word in or by within at the end of this sentence: ‘They also exhibit the fluidity, rapid transfer of heat and mass common in by liquids’. Only one word before liquids is needed.

Answer: Thank you for your careful review. We have made corresponding revisions in the manuscript.

Corresponding revision in Page 2:

They also exhibit the fluidity, rapid transfer of heat and mass common in liquids.

3. SCXRD analysis- The authors state that there are 8 Cl⁻ anions that are near the imidazolium groups. These Cl ions are shown in Figure S7, but not in Figure S8. Most importantly, they are not present in the supplied cif file. Moreover, the supplied cif also includes additional methylimidazolium molecules that sit in two of the four calixarene cavities. The source of the methylimidazolium is not clear, and it is not shown in the SI figures. If the supplied cif is incorrect, the correct file should be supplied to the CCDC.

Answer: Thank you very much for your careful inspection of our manuscript. In our previous manuscript, the Cl⁻ anions in Figure S8 were omitted, we have replaced the corresponding picture. The methylimidazolium parts are originated from the uncoordinated organic ligands. Because some atoms are not found, so we performed the squeeze operation. After double checking, there are indeed some errors in the CIF file. We re-provided the checked CIF file, and re-uploaded it to CCDC, the new CCDC number is 2259420.

Figure S8. Chemical structure of a cage in Zn-Cage. H atoms are omitted for clarity.

Table S1. Crystal data and structure refinement for Zn-Cage.

Identification code	Zn-Cage
CCDC number	2259420

4. The TGA of the Zn cage shows a slight mass loss at the beginning (before 100 °C). The authors should comment on the source of the mass loss. Is it unbound methylimidazolium? This type of mass loss is not seen for the PEG cage.

Answer: Thank you for your careful review. The Zn-cage was obtained after centrifugation and washing with methanol, while the PEG cage was obtained by washing with diethyl ether. So, the slight mass loss at the beginning in the TGA test of Zn cage may be the evaporation of residual methanol.

Corresponding revision in Page 9:

TGA demonstrated that the Zn-Cage is thermally stable up to 400 °C (Figure 1d), and

the slight weight loss before 100 °C, which was mainly attributed to the evaporation of residual methanol.

5. The authors state that 'Im-PL-Cage is a viscous paste-like material at room temperature (Scheme 1), with a melting point of ~50 °C (Figure 4b), which makes it a liquid slightly above room temperature.' I would argue that the use of the word slightly is a bit exaggerated as 20-25 °C is room temperature. This is a low melting point solid. Moreover, DSC shows 58 °C as the melting point, which is not ~50, it would be ~60 if you are rounding.

Answer: Thank you for your helpful suggestion. We have modified the relevant description.

Corresponding revision in Page 10:

Im-PL-Cage is a viscous paste-like material at room temperature (Scheme 1), with a melting point of ~58 °C (Figure 4b), which makes it a liquid at readily accessible temperatures.

6. In the SI, the percent yields are only included for some of the synthetic steps but should be added for all.

Answer: Thank you for your professional advice. We have added the percent yields in all synthesis steps, and we have supplemented with NMR characterization data for all the new compounds as well as their corresponding colors.

Corresponding revision in SI:

2.1 Synthesis of [(5-Meim-1,3-H₂BDC)⁺(Cl⁻)]^{1,2}:

(1) Synthesis of dimethyl 5-(1H-imidazol-1-yl)-isophthalate

.....The product was purified by chromatography on silica gel with hexane/ethyl acetate (2/1) to yield dimethyl 5-(1H-imidazol-1-yl)-isophthalate as a yellow powder (2.362 g, 9 mmol, 18% yield). ¹H NMR (400 MHz, DMSO-*d*₆, ppm): δ = 8.26 (s, 2H),

7.97 (s, 1H), 7.39 (s, 1H), 7.27 (s, 1H), 4.00 (s, 6H).

(2) Synthesis of [(5-Meim-1,3-H₂BDC)⁺(Cl⁻)]

.....The obtained residue was rinsed with hexane/ethyl acetate (2/1) to give dimethyl 5-(3-methyl-imidazol-1-yl)-isophthalate as a brown powder (1.58 g, 3.9 mmol, 56% yield). ¹H NMR (400 MHz, DMSO-*d*₆, ppm): δ 9.81 (s, 1H), 8.38 (s, 1H), 8.20 (m, 3H), 7.85 (s, 1H), 3.88(s, 6H), 3.94 (s, 3H).

.....The mixture was cooled down to room temperature, and the solvent was removed under reduced pressure to result [(5-Meim-1,3-H₂BDC)⁺(Cl⁻)] in light yellow powder. (0.8 g, 2.84 mmol, 73% yield). ¹H NMR (400 MHz, DMSO-*d*₆, ppm): δ 9.95 (s, 1H), 8.58 (s, 1H), 8.50 (m, 3H), 7.97 (s, 1H), 3.94 (s, 3H).

2.2 Synthesis of PEG-imidazolium 1,3-benzenedicarboxylic acid (PEG-Im-BDC):

(1) Synthesis of iodinated-methoxypolyethylene glycols

.....The desired product 4-methylbenzenesulfonate-methoxypolyethylene glycols was obtained as a white wax/oil (17.38 g, 4.1 mmol, 41% yield). ¹H NMR (400 MHz, DMSO-*d*₆, ppm): δ 7.85 (m, 2H), 7.40 (m, 2H), 3.66 (m, 360H), 3.24 (s, 3H), 2.35 (s, 3H).

.....The product iodinated-methoxypolyethylene glycols was obtained as a yellow wax/oil (15.22 g, 3.7 mmol, 90% yield). ¹H NMR (400 MHz, DMSO-*d*₆, ppm): δ 3.62 (m, 360H), 3.24 (s, 3H).

(2) Synthesis of PEG-Im-BDC

.....The final product PEG-Im-BDC as a dark yellow wax/oil (1.37 g, 0.32 mmol, 13% yield) was obtained by recrystallization from the filtrate at 273K. ¹H NMR (400 MHz, DMSO-*d*₆, ppm): δ 13.87 (s, 2H), 9.99 (s, 1H), 8.54 (m, 4H), 8.03 (s, 1H), 3.51 (m, 360H), 3.24 (s, 3H).

2.3 Synthesis of *p*-*tert*-butylsulfonylcalix[4]arenes (H₄TBSC)³⁻⁵:

(1) Synthesis of *p*-*tert*-Butylthiacalix[4]arene

.....The precipitate was collected by filtration, recrystallized from chloroform and dried in vacuo (100 °C, 4 h) to give an essentially pure sample of *p*-tert-Butylthiacalix[4]arene as pink powder (37.9 g, 49% based on the *p*-tert-butylphenol). The mother liquor of the recrystallization was concentrated in vacuo, and chromatography of the residue on silica gel (hexane/CHCl₃ = 4:6) afforded additional *p*-tert-Butylthiacalix[4]arene (3.9 g, 5%), the combined yield of *p*-tert-Butylthiacalix[4]arene amounting to 54% yield (41.8 g, 0.058 mol). ¹H NMR (400 MHz, CDCl₃, ppm): δ 9.60 (s, 4H), 7.64 (s, 8H), and 1.22 (s, 36H).

(2) Synthesis of *p*-tert-Butylsulfonylcalix[4]arene

.....The produce was recrystallized from benzene-methanol and dried in vacuo (70 °C, 12 h) to give the off-white product of *p*-tert-Butylsulfonylcalix[4]arene (1.06 g, 1.25 mmol, 90.6% yield). ¹H NMR (400 MHz, CDCl₃, ppm): δ 7.99 (s, 8H), 1.25 (s, 36H).

Reviewers' Comments:

Reviewer #1:

Remarks to the Author:

The authors have responded adequately to the reviewer's comments and questions, and the manuscript is acceptable now.

Reviewer #2:

Remarks to the Author:

Key characterizations include ^1H and ^{13}C NMR for all unknown compounds with high-resolution mass data or elemental analysis. The elemental data that the authors provided are not basic elemental analyses. The authors need to provide carbon, nitrogen, and hydrogen ratio for theoretical and experimental values. Please see this: <https://www.nature.com/ncomms/submit/chemical-characterisation>. The revised manuscript requires more characterization for their materials.

Reviewer #3:

Remarks to the Author:

The authors have addressed all the comments from all three reviewers to an adequate level. I recommend publication in Nature Communications.

The point-to-point responses to the referees' comments:

Reviewer #1 (Remarks to the Author):

The authors have responded adequately to the reviewer's comments and questions, and the manuscript is acceptable now.

Response: Thank you for your positive and encouraged comment.

Reviewer: 2

Key characterizations include ^1H and ^{13}C NMR for all unknown compounds with high-resolution mass data or elemental analysis. The elemental data that the authors provided are not basic elemental analyses. The authors need to provide carbon, nitrogen, and hydrogen ratio for theoretical and experimental values. Please see this: <https://www.nature.com/ncomms/submit/chemical-characterisation>. The revised manuscript requires more characterization for their materials.

Response: Thank you for your careful review. We have supplemented the ^{13}C NMR, high-resolution mass data, and elemental analysis for all unknown compounds. In addition, we have supplemented the theoretical values for the elemental analysis of materials in Supplementary Table 2, to facilitate readers' reading and analysis.

Corresponding revision in SI:

2.1 Synthesis of [(5-Meim-1,3-H₂BDC)⁺(Cl⁻)]^{1,2}:

(1) Synthesis of dimethyl 5-(1H-imidazol-1-yl)-isophthalate

.....The product was purified by chromatography on silica gel with hexane/ethyl acetate (2/1) to yield dimethyl 5-(1H-imidazol-1-yl)-isophthalate as a yellow powder (2.362 g, 9 mmol, 18% yield). ^1H NMR (400 MHz, DMSO-*d*₆, ppm): δ = 8.71 (s, 1H), 8.26 (s, 2H), 7.97 (s, 1H), 7.39 (s, 1H), 7.27 (s, 1H), 4.00 (s, 6H). ^{13}C NMR (400 MHz, DMSO-*d*₆, ppm): δ = 167.4, 137.2, 136.5, 131.3, 130.4, 126.5, 124.8, 123.6, 50.0. HRMS(ESI) *m/z*: [M]⁺ calculated for C₁₃H₁₂N₂O₄, 260.0795; found, 260.0793. Elemental analysis (calculated, found for C₁₃H₁₂N₂O₄): C (60.03, 60.24), H (4.65, 4.57), N (10.76, 10.83).

(2) Synthesis of [(5-Meim-1,3-H₂BDC)⁺(Cl⁻)]

.....The obtained residue was rinsed with hexane/ethyl acetate (2/1) to give dimethyl 5-(3-methyl-imidazol-1-yl)-isophthalate as a brown powder (1.58 g, 3.9 mmol, 56% yield). ^1H NMR (400 MHz, DMSO- d_6 , ppm): δ 9.81 (s, 1H), 8.38 (s, 1H), 8.20 (m, 3H), 7.85 (s, 1H), 3.88(s, 6H), 3.94 (s, 3H). ^{13}C NMR (400 MHz, DMSO- d_6 , ppm): δ = 167.9, 138.5, 135.1, 133.9, 130.7, 128.6, 125.9, 124.6, 50.5, 34.2. HRMS(ESI) m/z: $[\text{M}]^+$ calculated for $\text{C}_{14}\text{H}_{15}\text{IN}_2\text{O}_4$, 402.0077; found, 402.0076. Elemental analysis (calculated, found for $\text{C}_{14}\text{H}_{15}\text{IN}_2\text{O}_4$): C (41.81, 41.69), H (3.76, 3.79), N (6.97, 6.93).

.....The mixture was cooled down to room temperature, and the solvent was removed under reduced pressure to result $[(5\text{-Meim-1,3-H}_2\text{BDC})^+(\text{Cl}^-)]$ in light yellow powder. (0.8 g, 2.84 mmol, 73% yield). ^1H NMR (400 MHz, DMSO- d_6 , ppm): δ 9.95 (s, 1H), 8.58 (s, 1H), 8.50 (m, 3H), 7.97 (s, 1H), 3.95 (s, 3H). ^{13}C NMR (400 MHz, DMSO- d_6 , ppm): δ = 172.5, 140.8, 136.2, 131.7, 131.2, 128.5, 123.7, 35.6. HRMS(ESI) m/z: $[\text{M}]^+$ calculated for $\text{C}_{12}\text{H}_{11}\text{ClN}_2\text{O}_4$, 282.0413; found, 282.0413. Elemental analysis (calculated, found for $\text{C}_{12}\text{H}_{11}\text{ClN}_2\text{O}_4$): C (50.99, 50.65), H (3.92, 3.88), N (9.91, 9.83).

2.2 Synthesis of PEG-imidazolium 1,3-benzenedicarboxylic acid (PEG-Im-BDC):

(1) Synthesis of iodinated-methoxypolyethylene glycols

.....The desired product 4-methylbenzenesulfonate-methoxypolyethylene glycols was obtained as a white wax/oil (17.38 g, 4.1 mmol, 41% yield). ^1H NMR (400 MHz, DMSO- d_6 , ppm): δ 7.85 (m, 2H), 7.40 (m, 2H), 3.70 (s, 2H), 3.66 (m, 358H), 3.24 (s, 3H), 2.35 (s, 3H). ^{13}C NMR (400 MHz, DMSO- d_6 , ppm): δ = 143.8, 131.7, 130.5, 127.9, 73.2, 70.9, 70.5, 69.7, 62.4, 53.6, 20.8. HRMS(ESI) m/z: $[\text{M}]^+$ calculated for $\text{CH}_3(\text{OC}_2\text{H}_4)_n\text{SO}_3\text{C}_7\text{H}_7$ ($n = 85\text{-}95$), 3930.5166, 3974.5692, 4018.6254, 4062.6816, 4106.7378, 4150.7940, 4194.8502, 4238.9064, 4282.9626, 4327.0188, 4371.0750; found, 3930.5168, 3974.5689, 4018.6257, 4062.6813, 4106.7379, 4150.7941, 4194.8501, 4238.9066, 4282.9625, 4327.0187, 4371.0751. Elemental analysis (calculated, found for $\text{CH}_3(\text{OC}_2\text{H}_4)_n\text{SO}_3\text{C}_7\text{H}_7$): C (54.40, 55.75), H (8.98, 8.92).

.....The product iodinated-methoxypolyethylene glycols was obtained as a yellow

wax/oil (15.22 g, 3.7 mmol, 90% yield). ^1H NMR (400 MHz, DMSO- d_6 , ppm): δ 3.62 (m, 358H), 3.31 (s, 2H), 3.24 (s, 3H). ^{13}C NMR (400 MHz, DMSO- d_6 , ppm): δ = 73.9, 73.2, 70.8, 70.4, 69.6, 53.7, 7.9. HRMS(ESI) m/z : $[\text{M}]^+$ calculated for $\text{CH}_3(\text{OC}_2\text{H}_4)_n\text{I}$ ($n = 85-95$), 3886.4158, 3930.4720, 3974.5282, 4018.5844, 4062.6406, 4106.6968, 4150.7530, 4194.8092, 4238.8654, 4282.9216, 4326.9778; found, 3886.4160, 3930.4721, 3974.5283, 4018.5846, 4062.6405, 4106.6970, 4150.7531, 4194.8094, 4238.8653, 4282.9217, 4326.9776. Elemental analysis (calculated, found for $\text{CH}_3(\text{OC}_2\text{H}_4)_n\text{I}$): C (52.94, 52.87), H (8.91, 8.84).

(2) Synthesis of PEG-Im-BDC

.....The final product PEG-Im-BDC as a dark yellow wax/oil (1.37 g, 0.32 mmol, 13% yield) was obtained by recrystallization from the filtrate at 273K. ^1H NMR (400 MHz, DMSO- d_6 , ppm): δ 13.87 (s, 2H), 9.99 (s, 1H), 8.54 (m, 4H), 8.03 (s, 1H), 3.51 (m, 360H), 3.24 (s, 3H). ^{13}C NMR (400 MHz, DMSO- d_6 , ppm): δ = 172.1, 140.7, 136.5, 131.6, 131.3, 127.8, 127.5, 124.3, 80.2, 73.1, 70.7, 68.6, 53.8. HRMS(ESI) m/z : $[\text{M}]^+$ calculated for $\text{C}_{11}\text{H}_8\text{N}_2\text{O}_4\text{Cl}(\text{OC}_2\text{H}_4)_n\text{CH}_3$ ($n = 85-95$), 4024.1196, 4068.1758, 4112.2320, 4156.2882, 4200.3445, 4244.4006, 4288.4568, 4332.5130, 4376.5692, 4420.6254, 4464.6816; found, 4024.1195, 4068.1756, 4112.2322, 4156.2885, 4200.3447, 4244.4008, 4288.4566, 4332.5131, 4376.5693, 4420.6255, 4464.6817. Elemental analysis (calculated, found for $\text{C}_{11}\text{H}_8\text{N}_2\text{O}_4\text{Cl}(\text{OC}_2\text{H}_4)_n\text{CH}_3$): C (54.29, 54.97), H (8.80, 8.53), N (0.66, 0.75).

2.3 Synthesis of *p*-tert-butylsulfonylcalix[4]arenes (H_4TBSC)³⁻⁵:

(1) Synthesis of *p*-tert-Butylthiacalix[4]arene

.....The precipitate was collected by filtration, recrystallized from chloroform and dried in vacuo (100 °C, 4 h) to give an essentially pure sample of *p*-tert-Butylthiacalix[4]arene as pink powder (37.9 g, 49% based on the *p*-tert-butylphenol). The mother liquor of the recrystallization was concentrated in vacuo, and chromatography of the residue on silica gel (hexane/ CHCl_3 = 4:6) afforded additional *p*-tert-Butylthiacalix[4]arene (3.9 g, 5%), the combined yield of

p-tert-Butylthiacalix[4]arene amounting to 54% yield (41.8 g, 0.058 mol). ^1H NMR (400 MHz, CDCl_3 , ppm): δ 9.60 (s, 4H), 7.64 (s, 8H), and 1.22 (s, 36H). ^{13}C NMR (400 MHz, CDCl_3 , ppm): δ = 155.6, 143.7, 136.4, 120.5, 34.2, 31.3. HRMS(ESI) m/z : $[\text{M}]^+$ calculated for $\text{C}_{40}\text{H}_{48}\text{O}_4\text{S}_4$, 720.2135; found, 720.2136. Elemental analysis (calculated, found for $\text{C}_{40}\text{H}_{48}\text{O}_4\text{S}_4$): C (66.63, 66.87), H (6.71, 6.58).

(2) Synthesis of *p*-tert-Butylsulfonylcalix[4]arene

.....The produce was recrystallized from benzene-methanol and dried in vacuo (70 °C, 12 h) to give the off-white product of *p*-tert-Butylsulfonylcalix[4]arene (1.06 g, 1.25 mmol, 90.6% yield). ^1H NMR (400 MHz, CDCl_3 , ppm): δ 7.99 (s, 8H), 1.25 (s, 36H). ^{13}C NMR (400 MHz, CDCl_3 , ppm): δ = 153.2, 141.7, 132.6, 124.3, 34.8, 31.5. HRMS(ESI) m/z : $[\text{M}]^+$ calculated for $\text{C}_{40}\text{H}_{48}\text{O}_{12}\text{S}_4$, 848.2029; found, 848.2025. Elemental analysis (calculated, found for $\text{C}_{40}\text{H}_{48}\text{O}_{12}\text{S}_4$): C (56.58, 56.13), H (5.71, 5.89).

2.4 Synthesis of metal-organic cages:

(1) **Zn-Cage:**Yellow crystals of Zn-Cage that formed in 5 days were isolated by washing with methanol and dried in the air to give 63.5 mg of the as-synthesized material. ^1H NMR (400 MHz, $\text{DMSO-}d_6$, ppm): δ 9.37 (s, 1H), 8.46 (s, 1H), 8.23 (m, 3H), 8.13 (s, 1H), 7.94 (m, 4H), 3.90 (s, 3H), 1.22 (s, 18H). ^{13}C NMR (400 MHz, $\text{DMSO-}d_6$, ppm): δ = 175.3, 151.8, 140.9, 136.7, 132.4, 128.9, 124.1, 35.2, 31.6. HRMS(ESI) m/z : $[\text{M}]^+$ calculated for Zn-Cage, 1080.1473 1352.1285, 1689.0412, 2788.1227; found, 1080.1472 1352.1286, 1689.0410, 2788.1226. Elemental analysis (calculated, found for Zn-Cage): C (45.31, 44.87), H (3.42, 3.75), N (3.37, 3.56).

(2) **Im-PL-Cage:**The resulting oil was concentrated under reduced pressure yielding a dark yellow, viscous liquid Im-PL-Cage (378 mg). ^1H NMR (400 MHz, $\text{DMSO-}d_6$, ppm): δ 9.76 (s, 1H), 8.34 (m, 4H), 7.92 (m, 5H), 3.48 (m, 360H), 3.15 (s,

3H), 1.21 (s, 18H). ¹³C NMR (400 MHz, DMSO-*d*₆, ppm): δ = 170.8, 151.6, 140.7, 134.8, 131.5, 125.0, 122.7, 78.6, 72.3, 69.6, 53.8, 35.2, 31.5. HRMS(ESI) m/z: [M]⁺ calculated for Im-PL-Cage, 1197.1918, 1827.2885, 2503.9657, 3077.5915; found, 1197.1914, 1827.2886, 2503.9659, 3077.5916. Elemental analysis (calculated, found for Im-PL-Cage): C (52.80, 53.23), H (8.07, 8.05), N (0.58, 0.62).

Table S2. ICP and EA of Zn-Cage and Im-PL-Cage.

		Zn	C	H	N	S
Zn-Cage	ICP	15.73 %				
(Calculated)	EA		45.31 %	3.42 %	3.37 %	7.71 %
Zn-Cage	ICP	14.36 %				
(Found)	EA		44.87 %	3.75 %	3.56 %	7.81 %
Im-PL-Cage	ICP	2.72 %				
(Calculated)	EA		52.80 %	8.07 %	0.58 %	1.33 %
Im-PL-Cage	ICP	2.67 %				
(Found)	EA		53.23 %	8.05 %	0.62 %	1.35 %

Reviewer #3 (Remarks to the Author):

The authors have addressed all the comments from all three reviewers to an adequate level. I recommend publication in Nature Communications.

Response: Thanks for your positive and encouraged comment.